



# Radiative transfer acceleration based on the Principal Component Analysis and Look-Up Table of corrections: Optimization and application to UV ozone profile retrievals

Juseon Bak[a#] , Xiong Liu[a], Robert Spurr[b], Kai Yang[c],

Caroline R. Nowlan[a], Christopher Chan Miller[a], Gonzalo Gonzalez Abad[a],

and Kelly Chance[a]

[a]*Center for Astrophysics | Harvard & Smithsonian, Cambridge, MA, USA*

[b]*RT Solutions Inc., Cambridge, MA, USA*

[c]*Department of Atmospheric and Oceanic Science, University of Maryland College Park, College Park, Maryland, USA*

*#Currently at Pusan National University, Busan, Korea*

**Abstract**

In this work, we apply a principal component analysis (PCA)-based approach combined with look-up tables (LUTs) of corrections to accelerate the VLIDORT radiative transfer (RT) model used in the retrieval of ozone profiles from backscattered ultraviolet (UV) measurements by the Ozone Monitoring Instrument (OMI). The spectral binning scheme, which determines the accuracy and efficiency of the PCA–RT performance, is thoroughly optimized over the spectral range 265 to 360 nm with the assumption of a Rayleigh-scattering atmosphere above a Lambertian surface. The high level of accuracy (~ 0.03 %) is achieved from fast-PCA calculations of full radiances. In this approach, computationally expensive full multiple scattering (MS) calculations are limited to a small set of PCA-derived optical states, while fast single scattering and 2-stream multiple scattering calculations are performed, for every spectral point. The number of calls to the full MS model is only 51 in the application to OMI ozone profile retrievals with the fitting window of 270-330 nm where the RT model should be called at fine intervals (~0.03 nm with ~ 2000 wavelengths) to simulate OMI native measurements at 229 wavelengths (spectral resolution: 0.4-0.6 nm). We also developed a Look Up Table (LUT) to correct RT approximations performed using a scalar RT model with 4 streams (discrete ordinates) and 24 layers, thereby achieving the accuracy at the level attainable from simulations with a vector model with 12 streams and 72 layers; this speeds up the RT calculations by more than 2 orders of magnitude when ignoring other overhead. Overall, we speed up our OMI retrieval by a factor of 3.3 over the previous version, which has already been significantly sped up over line-by-line calculations due to various RT approximations. Improved treatments for RT approximation errors using LUT corrections improve



spectral fitting (2-5 %) and hence retrieval errors, especially for tropospheric ozone by up to ~10%; the
remaining errors due to the forward model errors are within 5 % in the troposphere and 3 % in the
stratosphere.

## 1. Introduction

Optimal estimation-based inversions have become standard for the retrieval of atmospheric ozone
profiles from atmospheric chemistry UV-Vis backscatter instruments. This inversion model requires
iterative simulations of not only radiances, but also of Jacobians with respect to atmospheric and surface
variables, until the simulated radiances are sufficiently matched with the measured radiances. These
ozone profile algorithms face a computational challenge for use in global processing of high
spatial/temporal resolution satellite measurements, due to on-line radiative transfer (RT) computations
at many spectral points from 270 to 330 nm; it is computationally very expensive to perform full
multiple-scattering (MS) simulations with the polarized RT model. To reduce the computational cost, a
scalar RT model can be applied together with a polarization correction scheme based on a LUT (Kroon
et al., 2011; Miles et al., 2015). Another approach is to carry out on-line vector calculations at a few
wavelengths (Liu et al., 2010) together with other approximations (e.g., low-stream, coarse vertical
layering, Lambertian reflectance for surface and cloud, no aerosol treatment). However, the
computational speed is still insufficient to process one day of measurements from the Aura Ozone
Monitoring Instrument (OMI) within 24 h (30 cross-track pixels $\times$ 1644 along-track pixels $\times$ 14
orbits) with reasonable computational resources. Consequently, only 20 % of the available OMI pixels
are processed to generate the operational ozone profile (OMO3PR) product (Kroon et al., 2011), and
the spatial resolution is degraded by a factor of 4 to produce the research ozone profile (OMPROFOZ)
product (Liu et al., 2010). With the advent of sophisticated inversion techniques and superior
spaceborne remote sensing instruments, computational budgets have increased rapidly in recent years.
Joint retrievals combining UV and thermal infrared ($\sim$ 9.6 $u$m) have been investigated to better
distinguish between upper- and lower-tropospheric ozone abundances from multiple instruments, e.g.,
OMI + TES, OMI +AIRS, and GOME-2 + IASI (Fu et al., 2013; 2018; Cuesta et al., 2013). The
geostationary satellite instrument Tropospheric Emissions: Monitoring of Pollution (TEMPO),
scheduled for Launch in 2022, is specially designed for joint retrievals combining UV and visible (540-
740 nm) radiances to enhance the performance of retrievals for ground-level ozone (Zoogman et al.,
2017). Moreover, the temporal and spatial resolutions of upcoming geostationary satellite instruments
are being improved, leading to a tremendous increase in the data volume to be processed; for example,
daily measurements of TEMPO (with ~2000 N/S cross-track pixels $\times$ ~1200 E/W mirror steps $\times$ ~8


times a day) are ~30 times greater in volume than those of OMI. Therefore, accelerating RT simulations
is one of the highest priority tasks to assure operational capability. For speed-up, LUTs have often been
used in trace gas retrieval algorithms to serve as proxies for RT modeling or to perform corrections to
on-line RT approximations. In recent years, applying neural network techniques and principal
component analysis (PCA) to RT computational performance has received quite a lot of attention (e.g.,
Natraj et al., 2005; Spurr et al., 2013;2016; Liu et al., 2016; Yang et al., 2016; Loyola et al., 2018; Nanda
et al., 2019; Liu et al., 2020).
The goal of this paper is to improve both computational efficiency and accuracy of RT
simulations in the OMI ozone profile algorithm (Liu et al., 2010) by combining a fast PCA-based RT
model with two kinds of correction techniques. The application of PCA to RT simulations was first
proposed by Natraj et al. (2005) by demonstrating a computational improvement of intensity simulation
in the $O_2 A$ band by a factor of 10 and with ~ 0.3 % accuracy compared to full line-by-line (LBL)
calculations. This scheme has been deployed to the UV-backscatter, thermal emission, and cross-over
régimes, and has been extended for the derivation of analytic Jacobians, for vector RT applications, and
for bidirectional surface reflectances (Kopparla et al., 2016; 2017; Natraj et al., 2010; Somkuti et al.,
2017; Spurr et al., 2013). The RT performance enhancement arises from a reduction in the number of
expensive full multiple scattering calculations; the PCA scheme uses spectral binning of the
wavelengths into several bins based on the similarity of their optical properties and the projection to
every spectral point of these full MS calculations which are executed for a small number of PCA-
derived optical states. In addition to the adaption of a PCA-based RTM for our ozone profile retrieval,
we have adopted the undersampling correction from our previous implementation (Kim et al., 2009;
Bak et al., 2019); this enables us to use fewer wavelengths for further speed-up without much loss of
accuracy. Furthermore, we have developed a LUT-based correction to accelerate on-line RT simulations,
by starting with a lower-accuracy configuration (scalar RT with no polarization, 4 streams, 24 layers)
and then correcting the accuracy to the level attainable by means of a computationally more expensive
configuration (vector RT, 12 streams, 72 layers). The stream value refers to the number of discrete
ordinates in the full polar space; thus, for example, the term "12 streams" indicates the use of 6
upwelling and 6 downwelling polar cosine discrete ordinate directions. In previous work, PCA-based
RT calculations were assessed mostly against LBL calculations, independently from the inverse model.
Therefore, the PCA performance is likely to be overestimated in terms of operational capability, because
operational algorithms have their own speed-up strategies with many approximations; this is the case
for our ozone profile algorithm. As mentioned above, the PCA-based RT model is employed in this
work to make forward-model simulations of OMI measurements for the retrieval of ozone profiles.



Therefore, we evaluate the operational capability of our retrieval algorithm in terms of the retrieval
efficiency as well as the accuracy, and assess these relative to the current operational implementation.

104        This paper is structured as follows. Section 2 describes the current forward model scheme and

evaluates the approximations made in RT calculations, with the determination of the configuration
parameters for accurate simulations. The updated forward model scheme is introduced for the PCA-
based RT model in Section 3.1, and the two kinds of correction schemes to use less spectral sampling
and less accurate RT configuration are detailed in Section 3.2. The evaluation is performed in Section
4 and then we summarize and discuss the results in Section 5.
**2. Current forward model scheme based on Vector LIDORT (VLIDORT) only**

111        We first describe the current v1 SAO OMI ozone profile algorithm that was implemented in OMI

Science Investigator-led Processing Systems (SIPS) to generate the research OMPROFOZ ozone
profile    product,    publicly    available    at    the    Aura    Validation    Data    Center    (AVDC,
https://avdc.gsfc.nasa.gov/index.php?site=1620829979&id=74). It employs the OMI UV channel that
is divided into UV1 (270-310 nm) and UV2 (310-380 nm). The spatial resolution of UV1 is degraded
by a factor of 2 in order to increase the signal to noise ratio (SNR) in this spectral region. The full width
at half maximum (FWHM) of the instrument spectral response function (ISRF) is ~0.63 nm for UV1
and ~0.42 nm for UV2, with corresponding spectral intervals of 0.33 nm and 0.14 nm, respectively. The
total number of OMI wavelengths used in our spectral fitting for ozone profiles is 229, from 270-308
nm (UV1) and 312-330 nm (UV2). The RT model needs to simulate sun-normalized radiances as well
as their derivatives with respect to the ozone profile elements and surface albedo. This simulation is
iteratively performed to ingest the atmospheric and surface variables adjusted through the physical
fitting between measured and simulated spectra and simultaneously the statistical fitting between the
state vector and the *a priori* vector. The retrieval is optimized within typically 2-3 iterations (up to 10
is permitted). The vertical grids of the retrieved ozone profiles in 24 layers are initially spaced in log
(pressure) at $P_i = 2^{-\frac{i}{2}} atm$ for 0 (surface), 23 (~55 km) and with the top of atmosphere set for $P_{24}$
(~65 km). Each layer is thus approximately 2.5-km thick, except for the top layer (~ 10 km). A number
of RT approximations have already been applied in the current forward model to speed up the processing.
In the remainder of this section, the current forward model scheme is described, with its flow chart
depicted in the left panel of Fig. 1. An error analysis is performed for optimizing the RT model
configuration to maximize the simulation accuracy.


In the first step, we select 93 effective wavelengths with variable sampling intervals, 1.0 nm below
295 nm, 0.4 nm from 295-310 nm, and 0.6 nm above 310 nm. The number of the wavelengths is smaller
than the OMI native pixels (229 from 270-330 nm) by more than a factor of 2. The on-line radiative
transfer model is run to generate the full radiance spectrum (single + multiple scattering) at these
wavelengths in the scalar mode, with 8 streams and a Rayleigh atmosphere divided into 25 layers – a
grid that is similar to that for the retrieval, except for the top layer (~ 55 km to 65 km) which is further
divided into two layers. Note that the Vector Linearized Discrete Ordinate Radiative Transfer VLIDORT
model Version 2.8 (Spurr and Christi, 2019) is implemented in this study. In step 2, a polarization
correction is applied to the scalar calculations done in step 1 using the on-line vector calculation at
fourteen wavelengths (visually shown with the vertical lines in Fig. 3.a.2). In step 3 the simulation at
the effective wavelength grid is interpolated into 0.05 nm intervals with the undersampling correction,
and the result is finally interpolated/convolved into OMI native grids in step 4.
Figure 2.b shows approximation errors related to undersampling from 0.02 nm to 0.1 nm compared
to the simulated radiance at the sampling rate of the ozone cross sections (0.01 nm) (Fig. 2.a). This
illustrates that current forward model calculation has trivial errors (less than 0.01 %) except for 0.02 %
around 310 nm if there is no error after undersampling correction to 0.05 nm. The correction applied in
step 3 allows relaxation of the sampling rate without loss of the accuracy. This correction is based on
the adjustment of the radiance due to the difference of the optical depth profiles between fine ($\lambda_h$) and
coarse ($\lambda_c$) spectral grids assisted by application of the weighting functions ($\frac{dI}{dx}$) as follows:
$$I(\lambda_h)$$

$$= I(\lambda_c)$$

$$+ \sum_{l=1}^{N} \frac{\partial I\ (\lambda_c)}{\partial \Delta_l^{gas}} \ \left( \Delta_l^{gas}(\lambda_h) - \Delta_l^{gas}(\lambda_c) \right)$$

$$+ \frac{\partial I\ (\lambda_c)}{\partial \Delta_l^{ray}} \left( \Delta_l^{ray}(\lambda_h) - \Delta_l^{ray}(\lambda_c) \right), \qquad (1)$$

151

where $\Delta_l^{gas}$ and $\Delta_l^{ray}$ are the optical depth profiles for trace gas absorption and Rayleigh scattering,
$l = 1, \cdots N_L$ (the number of atmospheric layers). Figure 2.c demonstrates that the undersampling
correction works well for simulations at 0.2 nm intervals or less over the entire spectral range, but it
can cause large errors when the simulations are performed at intervals of 1.0 nm, 0.4, and 0.6 nm for
the spectral ranges, 270-295 nm, 295-310 nm, and 310-330 nm, respectively. Figure 3 shows the


approximations applied to on-line VLIDORT calculations, including (a.1) neglect of the polarization
effect; (b) use of 8 streams; and (c) use of a coarse 24-layer height grid. As we mentioned above, in the
v1 forward model the scalar model is used for all wavelengths, with the vector model at 14 wavelengths
for correcting the scalar simulations. However, Figure 3.a.2 illustrates that second order of polarization
correction errors (~0.2 %) could remain due to neglecting the dependence of polarization effects on the
fine structures of ozone absorption. Using 8 streams causes errors of ~ 0.05 % above 320 nm, whereas
using the 24 layers causes 1 % errors at shorter UV wavelengths. Based on the results shown in Fig 3,
we conclude that there is room for improving the simulation accuracy by increasing the number of
streams to 12, dividing the atmosphere into 72 layers and using more wavelengths in the polarization
correction.
### 3.   The improved forward model scheme based on PCA-VLIDORT
The right panel of Fig. 1.2 illustrates the flow chart of the new forward model scheme (v2) which
employs the PCA-based RT model to perform on-line scalar simulations using 4 streams and a 24-layer
atmosphere for RT performance enhancement (step 1) and two kinds of correction schemes for
accounting for approximation errors (steps 2 and 3). Section 3.1.1 gives an overview on how the PCA
tool is combined with the VLIDORT Version 2.8 model; full theoretical details may be found in Spurr
et al. (2016) and Kopparla et al. (2017). Here, our paper gives details on how the PCA-based RT
configuration is optimized for the application to UV ozone profile retrievals for maximizing the speed-
up in the section 3.1.2. Section 3.2 specifies the step 2 wherein the LUT-based correction is applied to
simulation errors due to the use of a scalar model, a smaller number of streams and coarser-resolution
vertical grid. In the step 3 the undersampling correction is adopted from the v1 implementation, but the
Rayleigh scattering term of the equation 1 is neglected for the speed up with trivial loss of accuracy.
### 3.1.1 General PCA procedure
The PCA-based RT process begins with a grouping of spectral points into several bins; atmospheric
profile optical properties within each bin are similar. PCA is a mathematical transformation that
converts a correlated mean-subtracted dataset into a series of principal components (PCs). To enhance
RT performance, PCA is used to compress a binned set of correlated optical profile data into a small set
of atmospheric profiles which capture the vast majority of the data variance within the bin. The layer
extinction optical thickness $\Delta_{ni}$ and the single scattering albedos $\omega_{ni}$ are generally subjected to PCA,
where $n$ and $i$ are indices for atmospheric layers ($n = 1, \cdots N_L$) and spectral points ($i = 1, \cdots N_S$),



respectively. For each bin, the optical profiles $\{\ln \Delta_{ni}, \ln \omega_{ni}\}$ is composed of $2N_L \times N_S$ matrix $G$ in
log-space ($G_{n,i} = \ln \Delta_{ni}$, $G_{n+N_N,i} = \ln \omega_{ni}$). The mean-removed $2N_L \times 2N_L$ covariance matrix $Y$ is
then:
$$Y = [G - \langle G \rangle]^{\mathrm{T}}[G - \langle G \rangle], (2)$$
where $\langle \rangle$ denotes a mean-value over all grid points in a bin. This covariance matrix $Y$ is decomposed
into eigenvalues $\rho_k$ and unit eigenvectors $X_k$ through solution of the eigenvalue problem $YX_k =$
$\rho_k X_k$. The scaled eigenvectors of the covariance matrix are defined as the empirical orthogonal
function (EOFs), $W_k = \sqrt{\rho_k} X_k$, where the index $k$ is ranked from 1 to $2N_L$ in descending order
staring with the largest eigenvalues. The principal components (PCs) are the projections of the original
data onto the eigenvectors, $P_k = \frac{1}{\sqrt{\rho_k}} G W_k$. The original data set can then be expanded in terms of the
mean value and a sum over all EOFs. As inputs to the RT simulation, the PCA-defined optical states are
defined as $F_o = \exp[\langle G \rangle]$ and $F_k^{\pm} = F_o \exp[\pm W_k]$, corresponding respectively to the mean value and
to positive and negative perturbations from the mean value by an amount equal to the magnitude of $k^{\mathrm{th}}$
EOF. Therefore, $\Delta_{n,i}$ and $\omega_{n,i}$ (i=1…$N_S$) are expressed as followings:
$$F_o = \begin{Bmatrix} \Delta_{n,o} \\ \omega_{n,o} \end{Bmatrix} \equiv \begin{Bmatrix} \exp[\frac{1}{N_s}\sum_{i=1}^{N_s} \ln \Delta_{ni}] \\ \exp[\frac{1}{N_s}\sum_{i=1}^{N_s} \ln \omega_{ni}] \end{Bmatrix}; \quad F_k^{\pm} = \begin{Bmatrix} \Delta_{n,\pm k} \\ \omega_{n,\pm k} \end{Bmatrix} \equiv \begin{Bmatrix} \Delta_{n,o} \exp[\pm W_{n,k}] \\ \omega_{n,o} \exp[\pm W_{n+N_L,k}] \end{Bmatrix}. \quad (3)$$
For those optical quantities not included in the PCA reduction but still required in the RT simulations,
the spectral mean values for the bin are assumed, as long as they have smooth monotonic spectral
dependency or else are constant over the bin range. In our application, the phase functions and phase
matrices for Rayleigh scattering are derived from bin-average values of the depolarization factor.
Surface Lambertian albedos are constant in the RT simulation, but the calculated radiance is later
adjusted to account for $1^{\mathrm{th}}$ order wavelength dependency using surface albedo weighting functions. For
larger bins, it is possible to include the depolarization ratio or the Lambertian albedo as additional
elements in the optical data set subject to PCA; this has been investigated in another context by Somkuti
et al. (2017).
In the PCA-based RT package, three independent RT models are combined in order to generate the
full scattering intensity field ($I_{\mathrm{Full}}$) at each spectral point $\lambda_i$ in a single bin



$$I_{full}(\lambda_i) \cong [I_{2s}(\lambda_i) + I_{FO}(\lambda_i)]C(\lambda_i). \quad (4)$$

Two fast RT models, the "First-Order" (FO) and 2STREAM (2S), are used to generate an accurate
single scatter (SS) field ($I_{FO}$) and an approximate multiple scatter (MS) field ($I_{2S}$), respectively, for
every spectral point. The scalar 2S model computes the radiation field with 2 discrete ordinates only.
To derive the correction factors $C(\lambda_i)$, we first compute (logarithmic) ratios of the full-scatter and 2S-
based intensity fields calculated with PCA-derived optical states $F_o$ and $F_k^{\pm}$:
$$J_o = \ln\left[\frac{I_{VLD}(F_o) + I_{FO}(F_o)}{I_{2s}(F_o) + I_{FO}(F_o)}\right] \quad ; J_k^{\pm} = \left[\frac{I_{VLD}(F_k^{\pm}) + I_{FO}(F_k^{\pm})}{I_{2s}(F_k^{\pm}) + I_{FO}(F_k^{\pm})}\right]. \quad (5)$$

Intensity ratios at the original spectral points $J(\lambda_i)$ are then obtained using a second-order central
difference expansion based on the PCA principal components $P_{ki}$:
$$J(\lambda_i) = J_o + \sum_{k=1}^{N_{EOF}} \frac{(J_k^+ - J_k^-)}{2} P_{ki} + \frac{1}{2}\sum_{k=1}^{N_{EOF}} (J_k^+ - 2J_o + J_k^-)^2 P_{ki}^2. \quad (6)$$

The correction factors $C(\lambda_i) = \exp[J(\lambda_i)]$ are then applied to the approximate simulation $[I_{2s}(\lambda_i) +$
$I_{FO}(\lambda_i)]$ according to Equation 4 above. More details can be found in the literature (Natraj et al., 2005,
2010; Spurr et al., ;2013; 2016; Kopparla et al., 2017).
So far, we have discussed generation of total *intensity* field, using values $I_{FO}(\lambda_i)$ and $I_{2s}(\lambda_i)$
from full-spectrum FO and 2S model calculation, and PCA-derived values $I_{VLD}(F)$, $I_{2S}(F)$ and
$I_{FO}(F)$ based on PCA-derived optical states $F = \{F_o, F_k^{\pm}\}$. The above procedure works with
VLIDORT operating in scalar or vector mode; however, the 2S model is purely scalar, and cannot be
used if we want to establish PCA-RT approximations to the Q and U components of the Stokes vector
with polarization present. Instead, we rely on just the VLIDORT and FO models, and develop a PCA-
RT scheme based on the differences between the VLIDORT and FO Q/U values for monochromatic and
PCA-derived calculations, with an additive correction factor instead of the logarithmic ratios in
Equation (6) above. This was first introduced in Natraj et al. (2010), and is discussed in detail in Spurr
et al. (2016).
Of greater importance for us is the need to derive PCA-RT approximations to profile Jacobians
(weighting functions of the total intensity with respect to ozone profile optical depths). A PCA-RT





Jacobians scheme was developed by Spurr et al. (2013) for total column Jacobians in connection with
the retrieval of total ozone; this scheme involved formal differentiation of the entire PCA-RT system as
outlined above for the intensity field. This is satisfactory for bulk property Jacobians, but for profile
Jacobians it is easier to write (Efremenko et al., 2014; Spurr et al., 2016):
$$K_{Full}^{(\xi)}(\lambda_i) \cong \left[ K_{2S}^{(\xi)}(\lambda_i) + K_{FO}^{(\xi)}(\lambda_i) \right] D^{(\xi)}(\lambda_i), \qquad (7)$$

Here, $K_{2S}^{(\xi)}(\lambda_i) \equiv \frac{\partial I_{2S}(\lambda_i)}{\partial \xi}$, with similar definitions for the FO and VLIDORT partial derivatives. The
Jacobian correction factor $D^{(\xi)}(\lambda_i) = \exp[L^{(\xi)}(\lambda_i)]$ is determined using the same central-difference
expansion as that in Equation (6), but with quantities
$$L_0^{(\xi)} = \ln \left[ \frac{K_{VLD}^{(\xi)}(F_o) + K_{FO}^{(\xi)}(F_o)}{K_{2S}^{(\xi)}(F_o) + K_{FO}^{(\xi)}(F_o)} \right] \; ; L_{\pm k}^{(\xi)} = \left[ \frac{K_{VLD}^{(\xi)}(F_k^{\pm}) + K_{FO}^{(\xi)}(F_k^{\pm})}{K_{2S}^{(\xi)}(F_k^{\pm}) + K_{FO}^{(\xi)}(F_k^{\pm})} \right] \qquad (8)$$

in place of $J_o$ and $J_k^{\pm}$ in Equation (5).

### 3.1.2 The binning scheme

The major performance saving is achieved by limiting full-MS VLIDORT calculations to those
based on the reduced set of PCA-derived optical states $F_o$ and $F_k^{\pm}$. A general binning scheme has been
developed over the shortwave region from 0.29 to 3.0 μm (Kopparla et al. 2016), whereby the entire
region is divided into 33 specially-chosen sub-windows encompassing the major trace-gas absorption
signatures; in each such sub-window there are 11 bins for grouping optical properties, and up to four
EOFs for each PCA bin treatment; with this scheme, radiance accuracies of 0.1% can be achieved
throughout the region. However, the binning scheme should be tuned to the specific application to get
additional computational saving, and here, we investigate the optimal set for spectral binning and the
number of EOFs in the Hartley and Huggins ozone bands (265-360 nm).
Optical properties within each bin must be strongly correlated to reduce the number of EOFs
required to attain a given accuracy. According to Kopparla et al. (2016), the UV region is divided at 340
nm, beyond which $O_2$-$O_2$ absorption must be considered. In our application, the spectral region 340-
360 nm is further divided at 350 nm: in the first sub-window, ozone absorption is much stronger than
$O_2$-$O_2$, while for the second (350-360 nm), $O_2$-$O_2$ absorption becomes dominant. The binning criteria



are generally determined by similarities in total optical depth of gas absorption profiles $\tau_{ij}$ as defined
below:

$$\Gamma_g = -\ln \sum_{n=1}^{N_L} \sum_{j=1}^{N_g} \tau_{nj}, \qquad (9)$$

where $N_L$ and $N_g$ denote the number of atmospheric layers and atmospheric trace gases.

271        To evaluate the performance of the PCA approximation, the "exact-RT" model is executed in order

to calculate a fully accurate multiple-scattering spectrum using the FO model for an accurate single
scattering field and VLIDORT model for an accurate multiple scattering field:

$$I_{exact}(\lambda_i) = I_{VLD}(\lambda_i) + I_{FO}(\lambda_i). \qquad (10)$$

276        We first evaluate the impact of applying different binning steps and numbers of EOFs in Fig. 4.

where the residuals $(I_{PCA} - I_{EXACT})$ are plotted as a function of $\Gamma_g$ for the spectral window 265-340
nm at small and large SZAs, respectively. In this evaluation, the bins are equally spaced in $\Gamma_g$ for the
five steps from 0.20 to 1.0. For $\Gamma_g < 1$, where the extinction is strong enough that radiances are very
small, the residuals are effectively reduced by having more bins rather than increasing the number of
EOFs. In this optical range, using the first EOF is enough to capture the vast majority of the spectral
variance, with the optimization of the binning step. However, the bins should be narrowly spaced with
$\Gamma_g$ intervals of at least 0.3-0.4 for those spectral grids for which $\Gamma_g$ is less than -2. These spectral grids
are correlated with the Hartley band above ~300 nm, where radiance values rapidly increase due to
decreasing ozone absorption, but the spectral variations are almost unstructured. The rest of our
spectral region corresponds to the Huggins band above 310 nm, where spectral variations are distinctly
influenced by local maxima and minima of ozone absorption. In this spectral region, PCA
approximation errors can be greatly reduced by increasing the number of EOFs. However, it is
interesting to note that the PCA approximation is not further improved by using 4 EOFs instead of 3
(not shown here). Figure 4 also illustrates the dependence of the PCA performance on SZA in the
spectral range below 340 nm: For example, when 2 EOFs are applied with the binning step 0.4, errors
are within $\pm 0.02$ % at smaller SZA, but increase up to $\pm 0.03$ % at larger SZA. Therefore, as listed
in Table 1, two sets of binning criteria are determined to keep the accuracy within 0.05 % for any



viewing geometry. Based on the experiments shown in Fig. 5, the binning criteria are determined for
the other sub-windows listed in Table 1, namely 340-350 nm and 350-360 nm: The former is set with
bins at intervals of 1 and using the first two EOFs, while the latter is divided into a single bin with the
first four EOFs. Figure 6 illustrates the binning criteria thus determined, demonstrating that the PCA
performance keeps accuracies within 0.03 % when various sets of SZAs, ozone profiles, and vertical
layers are implemented.
**3.2 LUT-based correction**
Two sets of LUTs are created, for high accuracy (LUT$_H$: vector/12 streams/72 layers), and low
accuracy (LUT$_L$: scalar/4 streams/24 layers) configurations. The on-line PCA-VLIDORT model is
configured to run in the "LUT$_L$" mode. The correction spectrum is straightforwardly calculated as
the ratio of the LUT-based spectrum ($LUT_H/LUT_L$), but the radiance correction term is additionally
adjusted to account for the different gas optical depth profiles used in on-line and LUT simulations. The
RT results are corrected as follows.
$$I_{on} = I_{on,L} \times \exp\left( \ln(I_{LUT_H}/I_{LUT_L}) + \sum_{n=1}^{N_L} \left[ \left( \frac{\partial \ln I}{\partial \tau}_{LUT_H} - \frac{\partial \ln I}{\partial \tau}_{LUT_L} \right) \times (\tau_{on} - \tau_{LUT}) \right](n) \right); \quad (11a)$$

$$\frac{\partial I}{\partial A_{s_{on}}} = \frac{\partial I}{\partial A_{s_{on,L}}} \times \frac{\frac{\partial I}{\partial A_{s_{LUT_H}}}}{\frac{\partial I}{\partial A_{s_{LUT_L}}}}; \quad (11b)$$
$$\frac{\partial I}{\partial \tau_{on}} = \frac{\partial I}{\partial \tau_{on,L}} \times \frac{\partial I}{\partial \tau_{LUT_H}} \Big/ \frac{\partial I}{\partial \tau_{LUT_L}}, \quad (11c)$$
where $A_s$ and $\tau_n$ represent the surface albedos and gas absorption optical depths (n is the layer index).
To construct LUTs, RT calculations are performed using the VLIDORT version 2.8 model for sets of
geometrical configurations ($\theta, \theta_o$; solar zenith angle, viewing zenith angle), surface pressures for
22 climatological ozone profiles and 92 wavelengths (265-345 nm) as listed in Table 2. The azimuth
dependence is treated exactly using the 0-2 Fourier intensity components in a Rayleigh scattering
atmosphere in conjunction with the associated cosine-azimuth expansion of the full intensity; see the



discussion below. The 22 ozone profiles are constructed from the GOME ozone profile product (Liu
et al. 2005), where the ozone profile shapes vary according to three latitude regimes and with the total
column ozone amounts at 50 DU intervals. The 92 wavelengths are regularly sampled at 5 nm intervals
below 295 nm and at 1.0 nm intervals up to 310 nm in the Hartley band, while irregularly sampled
based on the local minima and maxima of the ozone absorption structures in the Huggins band. The
results of these RT calculations are separated into two components: the path radiance $I_{atm}$ and the
surface reflectance term $I_{sfc}$ according to Chandrasekhar (1960), so that the following relationship
is employed to recover the full radiance:
$$I(\theta, \theta_o, \varphi - \varphi_o, A_s) = I_{atm}(\theta, \theta_o, \varphi - \varphi_o) + I_{sfc}(\theta, \theta_o, A_s). \quad (12)$$


$I_{atm}$ represents the purely atmospheric contribution to the radiance in the presence of a dark surface
(zero albedo), and in a Rayleigh scattering atmosphere, this is given as a Fourier expansion in the cosine
of the relative azimuth angle.
$$I_{atm}(\theta, \theta_o, \varphi - \varphi_o) = I_o(\theta, \theta_o) + \cos(\varphi - \varphi_o) I_1(\theta, \theta_o) + \cos 2(\varphi - \varphi_o) I_2(\theta, \theta_o). \quad (13)$$

330

332 However, it is more convenient to write this in the form:

334
$$I_{atm} = I_o(\theta, \theta_o)\left(1 + aq_1 \cos(\varphi - \varphi_o) Z_1(\theta, \theta_o) + aq_2 \cos 2(\varphi - \varphi_o) Z_2(\theta, \theta_o)\right); \quad (14a)$$

333

335
$$Z_1(\theta, \theta_o) = \frac{1}{aq_1} \frac{I_1(\theta, \theta_o)}{I_0(\theta, \theta_o)}; \qquad Z_2(\theta, \theta_o) = \frac{1}{aq_2} \frac{I_2(\theta, \theta_o)}{I_0(\theta, \theta_o)}; \quad (14b)$$

337
$$aq_1 = \frac{3}{8} \cos\theta \sin\theta \sin\theta_o; \quad aq_2 = \frac{3}{32} \frac{(\sin\theta \sin\theta_o)^2}{\cos\theta_o}. \quad (14c)$$


In the LUTs, the three coefficients ($I_o$, $Z_1$, and $Z_2$) are stored instead of $I_{atm}$. Note that the use
of terms $aq_1$ and $aq_2$ is taken from Dave (1964); most of the angular variability in
components $I_1$ and $I_2$ are captured analytically with these functions. In other words, $Z_1$ and
$Z_2$ are angularly smooth and well-behaved (non-singular) functions, which helps improve





angular interpolation accuracy with fewer points in the angular grids. The surface term is

$$I_{sfc}(\theta, \theta_o, A_s) = \frac{A_s T(\theta, \theta_o)}{1 - A_s s^*}. \quad (15)$$


In the LUTs, we store the transmission term $T(\theta, \theta_o)$, which is the product of the atmosphere
downwelling flux transmittance for a solar source with the upwelling transmittance from a
surface illuminated isotropically from below, and the geometry-independent term $s^*$ which is
the spherical albedo from such a surface. This is the so-called "planetary problem" calculation
(Chandrasekhar, 1960), and the code to obtain T and $s^*$ is now implemented in VLIDORT
Version 2.8 (Spurr, 2019). One of the key features of the VLIDORT code is its ability to
generate simultaneously (along with the Stokes vector radiation field) any set of Jacobians with
respect to atmospheric and surface optical properties. VLIDORT also contains an analytical
linearization of the planetary problem. Indeed, in our Rayleigh-based application, we require
Jacobians with respect to the albedo $A_s$ and the ozone profile elements $\tau$. First for the albedo
weighting function we have straightforward differentiation from Equation (15) as following

$$\frac{\partial I}{\partial A_s} = T(\theta, \theta_o) \left(\frac{qr}{A_s}\right)^2; \qquad qr = A_s/(1 - A_s s^*). \quad (16)$$



For the optical depth derivative, $\partial I/\partial \tau$ is calculated from

$$\frac{\partial I}{\partial \tau} = \frac{\partial I_o}{\partial \tau} + aq_1 \cos(\varphi - \varphi_o) \frac{\partial Z_1}{\partial \tau} + aq_2 \cos 2(\varphi - \varphi_o) \frac{\partial Z_2}{\partial \tau} + qr.\frac{\partial T}{\partial \tau} + T.(qr)^2.\frac{\partial s^*}{\partial \tau}. \quad (17)$$



All partial derivatives in this expression are returned automatically by VLIDORT. For a given
ozone profile, wavelength, and surface pressure, the number of the LUT values specified in
Table 3 is 770 (nVar $\times$ n$\theta$ $\times$ n$\theta_o$ + $S_b$ + $\frac{dS_b}{d\tau}$, nVar = 8: $I_o$, $Z_1, Z_2, T, \frac{dI_o}{d\tau}, \frac{dZ_1}{d\tau}, \frac{dZ_2}{d\tau}, \frac{dT}{d\tau}$), which is
much smaller than that of a LUT with dependence on 8 relative azimuth angles and 5 surface
albedo values (11,520 =nVar $\times$ n$\theta$ $\times$ n$\theta_o$ $\times$ n$(\varphi - \varphi_o)$ $\times$ nA$_s$, nVar = 3: I, $\partial I/\partial \tau, \partial I/\partial A_s$). LUT-based
simulated radiances are evaluated against on-line simulations: The LUT interpolation errors



are mostly less than 0.2-0.3 % (not shown here), except for extreme path length scenarios (e.g.,
~1% at $\theta_o = 87.0°$) as shown in Fig. 7 a.b, However the interpolation errors are quite similar
to each other for $LUT_H$ and $LUT_L$. Therefore, those errors are canceled out when performing
corrections using the two LUTs and thereby the overall error after LUT correction is much
smaller than ~ 0.05 % (Fig. c). Note that the accuracy is completely maintained with respect to
both $\varphi - \varphi_o$ and $A_s$, while the size of a LUT is reduced by a factor of 15. However,    LUT
corrections still contain ozone profile shape errors due to the use of 22 representative total
ozone-dependent ozone profiles in the LUT. Figure 8 shows an example of the correction
spectrum as a function of SZA, showing that polarization errors are mostly dominant, except
at the high SZAs above 310 nm, where errors due to use of a low number of streams become
significant, and for wavelengths below 300 nm where the use of the coarse vertical layering
scheme becomes the main source of uncertainty.
4. **Evaluation**
The PCA-RT model developed as described in this paper is implemented as the forward model
component of an iterative OE based inversion for retrieving ozone profile from OMI measurements. In
previous studies, the PCA-RT performance was evaluated against a suite of exact monochromatic
baseline of fully accurate VLIDORT simulations. However, such exact RT calculations cannot be
applied in the operational data processing system, especially when thousands of spectral points are
involved; in other words, the operational capability of the PCA-RT approach has been overestimated in
previous studies. Therefore, we evaluate the RT model developed against the existing forward model
where many RT approximations are applied to meet the computational budget in the operational system.
Table 4 contains sets of configurations for 7 forward models. OMI spectra are simulated at the under-
sampled ("US") intervals specified in the first column of this table and then interpolated at high-
resolution ("HR") intervals (second column) with the undersampling correction before convolution with
OMI slit functions. In the v1 forward model, the US spectral intervals were set at 1.0 nm/0.4 nm
intervals below/above 295 nm and 0.6 nm above 310 nm, while the HR spectral interval was set at 0.05
nm. In the updated RT model, the spectral points are selected at 0.3 nm (0.1 nm) intervals below (above)
305 nm and the HR interval is set as 0.03 nm, which enables us to achieve very high-accuracy, better
than 0.01 %, as shown in Fig. 2.c. In the reference configuration (abbreviated to "Ref"), VLIDORT is
run in vector mode with 12 streams and 72 atmospheric layers, so that the RT approximation errors are


significantly eliminated. The VLIDORT-based forward model is run with five sets of configurations
(abbreviated to "VLD" in Table 4) to quantify the impact of RT approximations on ozone retrievals.
Figure 10 compares the mean biases of the retrieved ozone profiles between VLD/PCA and Ref for
three SZA regimes. VLD$^0$ represents the v1 forward model configuration, demonstrating that the ozone
retrieval errors due to the entire forward model errors range from ~ 2 % for the large SZA regime to ~
5 % for the small SZA regime at the lower atmospheric layers, but ~ 1 % at the upper layers. The
configuration VLD$^1$ assesses the impact of undersampling errors on the retrievals, causing negative
biases of up to 2.0 % below ~ 20 km. Compared to the use of 12 streams, using 8 streams causes
negligible impacts on ozone retrievals (VLD$^2$) as the corresponding RTM approximation errors are
negligible, except for extreme viewing geometries where the ozone retrieval errors are overwhelmed
by instrumental measurement errors (a few %), rather than the forward model errors of ~ 0.05 % as
shown in Fig. 3. The VLD$^3$ based RT calculation is applied to ozone retrievals for evaluating on-line
polarization correction, showing that the corresponding errors in tropospheric ozone retrievals are
estimated as $\pm$ 2 % at small SZAs. The evaluation for VLD$^4$ demonstrates that the use of coarse
atmospheric layering causes the largest errors (~4.5 % in the troposphere, ~ 1.5 % in the stratosphere).
PCA$^o$ represents the v2 forward model configuration while PCA$^1$ is done in the highest accurate
configuration except for PCA approximation. Retrieval errors due to PCA approximation are negligible
except for the bottom few layers at smaller solar zenith angles (up to ~ 1.5 %). Differences between
PCA$^o$ and PCA$^1$ represents the ozone retrieval errors due to LUT errors, mostly related to the profile
shape errors between LUT and on-line calculations. In Fig. 10, the comparison between VLD0 (v1
PROFOZ) and PCA0 (v2 PROFOZ) is performed for individual ozone profile retrievals. The large
systematic errors of ~ 5- 15 % due to v1 forward model errors are greatly eliminated below 30 km. In
addition, the random-noise errors are significantly eliminated over the entire layers at high solar zenith
angles. However, there are still some remaining retrieval errors up to - 5% in troposphere and 3 % in
stratosphere due to v2 forward model simulation errors. Figure 11 further evaluates the v2
implementation. First of all, the comparison of the runtime (Fig 11.a) demonstrates a 3.3-fold-increase
in speed on average, thanks to switching the forward model from v1 to v2. Some spectral fit residuals
are eliminated in the UV 1 band over the middle area of the swath (low latitudes), where the SZAs are
relatively small, by up to ~ 2 %; the corresponding improvements are found in the stratospheric column
ozone. The amount of the stratospheric column ozone deviated from the reference is reduced by ~ 0.2 %
with v2 implementation. On the other hand, the tropospheric column ozone retrievals show
improvements for most cases, whereas the fit residuals of the UV2 band are slightly worse in the low
latitudes, but significantly better (2-5 %) in the Northern high latitudes (OMI along-track number > ~



429     1300).

**5. Summary and Conclusions**
We have extended the PCA-based fast RT method to overcome computational challenges for OE-
based SAO OMI ozone profile retrievals from ultraviolet measurements requiring iterative calculations
of the radiance and its Jacobian derivatives, to match the simulated spectrum to the measured spectrum
The PCA-RT model is designed to perform MS calculations for a few EOF-derived optical states which
are developed from spectrally binned sets of inherent optical properties that possess some redundancy.
To maximize the performance enhancement, we carefully tuned the binning scheme for the UV ozone
fitting window from 265 nm to 360 nm in such a way as to choose the number of EOFs to be as small
as possible for each bin, rather than always using the first four EOFs for all bins selected in previous
studies. The spectral windows are divided into three sub-windows: 1) 265-340 nm, 2) 340-350, and 3)
350-360 nm. Then, optical profiles are grouped into bins according to criteria based on the total gas
optical depth, as specified in Table 1. Spectral bins correlated to the Hartley ozone band use only the
first EOF, but 2 or 3 EOFs are required for those bins related to the Huggins ozone band. The MS model
is executed 85 times for the entire wavelength range (265-360 nm), and only 51 times for the OMI
ozone fitting window (270-330 nm). We demonstrated that the PCA approximation errors are within
0.03 % for any viewing geometry, optical depth profile, and vertical layering. The existing (v1) forward
model calculations are evaluated to determine the optimal configuration for the v2 forward model. RT
approximation errors exist due to the use of 24 quite coarse vertical layers (2.5 km thick), which can
cause radiance simulation errors of up to ~ 1 % below 320 nm and this leads to ozone retrieval errors of 2-
4 % in the troposphere and 1 % in the stratosphere. Eight-stream calculations can result in radiance residuals
of ~ 0.05 % or less except at extreme viewing geometries, which causes trivial errors on ozone retrievals
compared to other error factors. In spite of accounting for polarization errors using vector and scalar
differences at 14 wavelengths, the retrieval accuracies are systematically worse by ~ 2 % due to neglecting
second-order polarization errors which are strongly correlated with ozone absorption features. We found that
72 atmospheric layers (~ 0.7 km thick) and 12 streams should be used at least for fully accurate RT
calculations comparable to those with 99 atmospheric layers and 32 streams. The OMI spectral fit uses 229
wavelengths at OMI native grids, but the existing RT model simulates 93 wavelengths and is then
interpolated onto 0.05 nm grids with undersampling correction. However, we found room to improve our
retrievals (~ 1.5 % on avearge) by simulating 244 wavelength grids selected at intervals of 0.3 nm/0.1 nm
below/above 305 nm and then performing the undersampling correction to 0.03 nm. Applying the PCA-RT
approach allows us to reduce the number of MS calculations from the high-resolution optical dataset to 51



sets of EOF-derived optical states, but the performance savings are not enough to improve over previous RT
approximations. To improve both efficiency and accuracy, we have developed a LUT-based correction for
eliminating the RT approximation errors arising from the vector vs scalar, 12 vs. 4 streams, and 72 vs. 24
layers. In conclusion, the updated PCA-based RT model combined with LUT corrections makes ozone
profile retrievals faster than the v1 forward model by a factor of 3.3 on average. Improvements in fitting
accuracies are also achieved in the UV1 band by 2 % and in the UV2 band by 2-5 %. Correspondingly,
the ozone profile retrievals are significantly improved, especially in the troposphere by ~ up to 10 %.
However, there are still some remaining retrieval errors of up to - 5% in troposphere and 3 % in
stratosphere due to the LUT correction errors and PCA approximation errors in the v2 implementation.
The updated forward model is in preparation for reprocessing all OMI measurements (2004 -current)
for the next version of the PROFOZ product.

*Author contributions*. JB and XL designed the research; RS provided oversight and guidance for using
both VLIDORT and PCA-based VLIDORT; KY developed the LUT creation and interpolation scheme;
XL contributed to analyzing ozone profile retrievals with different forward model approaches; JB
conducted the research and wrote the paper; CN, CM, GA, and KC contributed to the analysis and
writing; CM and GA contributed to managing the computational resources.
*Competing interests*. The authors declare that they have no conflicts of interest.
*Data availability*. OMI Level1b radiance datasets are available at
https://aura.gesdisc.eosdis.nasa.gov/data/Aura_OMI_Level1/ (last access: 31 AUG 2020). The LUT
database are attainable upon request.
*Acknowledgments* We acknowledge the OMI science team for providing their satellite data. Research
at the Smithsonian Astrophysical Observatory is funded by the NASA Aura science team program
(NNX17AI82G). Research at Pusan National University is funded by Basic Science Research Program
through the National Research Foundation of Korea(NRF) funded by the Ministry of
Education(2020R1A6A1A03044834).
*Financial support*. This research has been supported by NASA Aura science team program (grant no.
NNX17AI82G) and Basic Science Research Program through the National Research Foundation of
Korea(NRF) funded by the Ministry of Education(2020R1A6A1A03044834).



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





**Table 1. The PCA-RT configuration optimized over the UV spectral range 265-360 nm. The**
**optical depth of the total gas column ($\Gamma_g$ defined in eq. 9) is used to set the criteria for the spectral**
**binning; for example, one or more bins are created at intervals ($\Delta\Gamma_g$) in the range $\Gamma_g^{min}$ to $\Gamma_g^{max}$.**
**For each bin, the optical states are expanded in terms of the first few number of EOFs (nEOF).**

| 265-340 nm | | | | | | | |
|---|---|---|---|---|---|---|---|
| SZA or VZA < 70 º | | | | SZA or VZA $\geq$ 70º | | | |
| List | $\Gamma_g^{lower}$, $\Gamma_g^{upper}$ | $\Delta\Gamma_g$ | nEOF | List | $\Gamma_g^{lower}$, $\Gamma_g^{upper}$ | $\Delta\Gamma_g$ | nEOF |
| 1 | $\infty$ to -1.7 | 2 | 1 | 1 | $\infty$ to - 1.5 | 2.0 | 1 |
| 2 | -1.7 to -1.2 | 0.5 | 1 | 2 | -1.5 to -0.7 | 1.2 | 1 |
| 3 | -1.2 to 0.0 | 0.4 | 1 | 3 | -0.7 to 0.4 | 0.35 | 1 |
| 4 | 0.0 to 0.5 | 0.5 | 1 | 4 | 0.4 to 0.7 | 0.3 | 1 |
| 5 | 0.5 to 3.5 | 0.6 | 2 | 5 | 0.7 to 2.5 | 0.6 | 3 |
| 6 | 3.5 to 4.5 | 1.0 | 2 | 6 | 2.5 to 3.5 | 1.0 | 3 |
| 7 | 4.5 to $\infty$ | 2.0 | 2 | 7 | 3.5 to 4.5 | 1.0 | 2 |
| 8 | | | | 8 | 4.5 to $\infty$ | 2.0 | 2 |
| 340-350 nm | | | | 350-360 nm | | | |
| List | $\Gamma_g^{lower}$, $\Gamma_g^{upper}$ | $\Delta\Gamma_g$ | nEOF | List | $\Gamma_g^{lower}$, $\Gamma_g^{upper}$ | $\Delta\Gamma_g$ | nEOF |
| 1 | $\infty$ to $\infty$ | 1.0 | 2 | 1 | $\infty$ to $\infty$ | $\infty$ | 4 |

**Table 2. LUT parameter specification. Note that the relative azimuth dependence is taken**
**into account explicitly through the Fourier coefficients of path radiance (Table 3) and the**
**surface albedo dependence is taken into account by the planetary problem.**

| Parameter | Symbol | N | Grid Values |
|---|---|---|---|
| Ozone Profile[+] | $O_3P$ | 22 | • Low-latitude (30ºS-30ºN)<br>  L200,L250,L300,L350<br>• Mid-latitude (30º-60ºN/S)<br>  M200,M250,M300,M350, M400,M450, M500,M550<br>• High-latitude (60º-90ºN/S)<br>  H100,H150,H200,H250,H300,H350, H400,H450,<br>  H500,H550 |
| Wavelength | $\lambda$ | 92 | 265-345 nm |
| Solar Zenith Angle | $\theta_o$, | 12 | 0, 16, 31, 44, 55, 64, 71, 76.5, 80.5, 83.5, 86, 88° |
| Viewing Zenith Angle | $\theta$ | 8 | 0, 15, 30, 43, 53, 61, 67, 72° |
| Surface albedo | $A_s$ | 1 | 0.0 |
| Surface pressure | $P_s$ | 12 | 100, 150, 200, 300, 400, 500, 600,<br>700, 800, 900, 1013.25, 1050 hPa |

[+]Total ozone-based ozone profiles for three latitude regimes. The grid values represent the
amount of total ozone (DU).
**Table 3. LUT variable specification**

| Variable | Dimensions | Variable | Dimensions |
|---|---|---|---|
| $I_o$[a] | $n\lambda, n\theta, n\theta_o, nP_s$ | $dI_o/d\tau$ | $n\lambda, n\theta, n\theta_o, nz, nP_s$ |
| $Z_1$[a] | $n\lambda, n\theta, n\theta_o, nP_s$ | $dZ_1 d\tau$ | $n\lambda, n\theta, n\theta_o, nz, \ nP_s$ |
| $Z_2$[a] | $n\lambda, n\theta, n\theta_o, nP_s$ | $dZ_2/d\tau$ | $n\lambda, n\theta, n\theta_o, nz, nP_s$ |
| $T$[b] | $n\lambda, n\theta, n\theta_o, nP_s$ | $dT/d\tau$ | $n\lambda, n\theta, n\theta_o, nz, nP_s$ |
| $S_b$[c] | $n\lambda, nP_s$ | $dS_b/d\tau$ | $n\lambda, nz, nP_s$ |
| $\tau$[d] | $n\lambda, nz^+$ |  |  |

[a] Fourier coefficients of path radiance with respect to relative azimuth angle
[b] total transmission of the atmosphere
[c] spherical albedo of the atmosphere
[d] total gas absorption optical depth profile
[+] Number of atmospheric layers


**Table 4. List of configurations used in evaluating the different forward model**
**calculations for OMI ozone profile retrievals. The reference, VLIDORT, and PCA-RT**
**models are abbreviated as Ref, VLD, and PCA, respectively.**

| RT models | US SI (nm) [a] | HR SI (nm) [b] | Nstream[c] | Nlayer[d] | Polarization[e] | RT corr[f] |
|---|---|---|---|---|---|---|
| **Ref** | 0.3(<305nm) 0.1 (≥305 nm) | 0.03 | 12 | 72 | True | False |
| **VLD[o]** | 1.0(<295nm) 0.4(between) 0.6 (≥310 nm) | 0.05 | 8 | 24 | False | On-line polcorr |
| **VLD[1]** | 1.0(<295nm) 0.4(between) 0.6 (≥310 nm) | 0.05 | 12 | 72 | True | False |
| **VLD[2]** | 0.3(<305nm) 0.1 (≥305 nm) | 0.03 | 8 | 72 | True | False |
| **VLD[3]** | 0.3(<305nm) 0.1 (≥305 nm) | 0.03 | 12 | 72 | False | On-line polcorr |
| **VLD[4]** | 0.3(<305nm) 0.1 (≥305 nm) | 0.03 | 12 | 24 | True | False |
| **PCA[0]** | 0.3(<305nm) 0.1 (≥305 nm) | 0.03 | 4 | 24 | False | LUT |
| **PCA[1]** | 0.3(<305nm) 0.1 (≥305 nm) | 0.03 | 12 | 72 | True | False |

[a] Under-sampled (US) spectral intervals (nm) used to define wavelengths at which RT is actually executed.
[b] High-resolution (HR) spectral intervals (nm) used to define wavelengths where under-sampled simulations are
interpolated before spectral convolution.



$^c$ the number of discrete ordinates in the full polar space; $^d$Number of atmospheric layers
$^e$RT model is run in the vector (scalar) mode if polarization is true (false).
$^f$On-line polarization correction as described in Section 2, which is originally developed from Liu et al. (2010).
LUT-based correction introduced in Section 3.3, which is developed in this work to account for RT approximation
errors due to neglecting polarization as well as using 4 streams and 24 layers.

**V1: VLIDORT**    **V2: PCA-VLIDORT**

| **STEP1: N($\lambda_e$) = 93** | **STEP1: N($\lambda_e$) = 244** |
|---|---|
| $I_s$ ($\lambda_e$) = **RT**$_{VLD}$ (8st, 25 layers, scalar) | $I_s$ ($\lambda_e$) = **RT**$_{PCA}$ (4st, 24 layers, scalar) |

| **STEP1: on-line Polarization correction** | **STEP2: LUT correction** |
|---|---|
| $I_v$ ($\lambda_c$) = **RT**$_{VLD}$ (8st, 25 layers, vector) | $\Delta C$ ($\lambda_c$) = **LUT**$_H$/**LUT**$_L$ |
| $\Delta C$ ($\lambda_c$) = $I_v/I_s$ | $I = I_s \times \Delta C$ |
| $I = I_s \times \Delta C$ | |
| * N($\lambda_c$) = 14 | * N($\lambda_c$) = 64 |

| **STEP3: undersampling correction** | **STEP3: undersampling correction** |
|---|---|
| $I (\lambda_h) = I(\lambda_e) + \dfrac{dI}{dgas}\Delta_{gas} + \dfrac{dI}{dray}\Delta_{ray}$ | $I (\lambda_h) = I(\lambda_e) + \dfrac{dI}{dgas}\Delta_{gas}$ |
| * N($\lambda_h$) = 1160 @ 0.05 nm | * N($\lambda_h$) = 1934 @ 0.03 nm |

**STEP4: convolution and interpolation**

$$I(\lambda_{OMI}) = I(\lambda_h) \otimes S$$


**Fig.1**. Schematic flowcharts of VLIDORT (v1) and PCA-VLIDORT (v2) based forward
models, respectively. Note that VLIDORT was used in the generation of the OMPROFOZ v1
dataset, while PCA-VLIDORT is in preparation for OMPROFOZ v2 production. The number
of wavelengths used in each process is denoted as N($\lambda$) when the spectral window 270-330
nm is applied. $\lambda_e$ represents the wavelength grids used for RT calclation, while $\lambda_c$ and $\lambda_h$
are grids used in RT approximation correction and undersampling correction, respectively.
See text for definition of other variables.





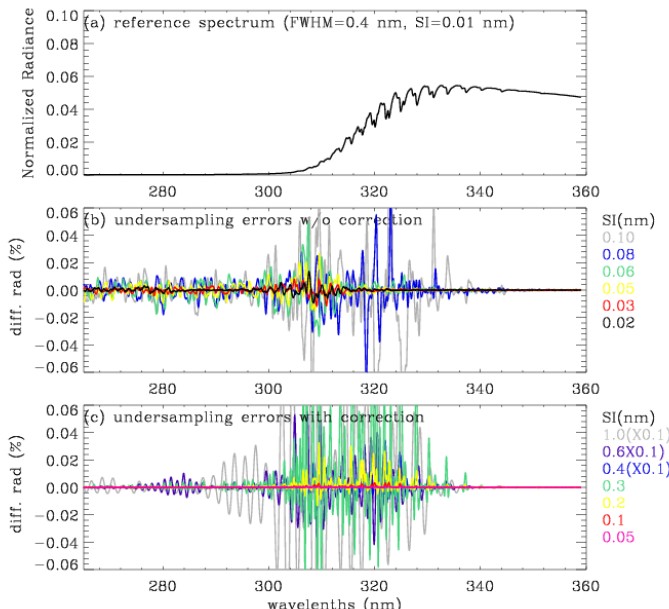


**Fig.2**. (a) Reference (truth) normalized radiance spectrum simulated at the spectral intervals (SIs) of 0.01 nm in 265-360 nm (solar zenith angle = 65 °, viewing zenith angle = 30 °, relative azimuth angle = 120º), which is used for evaluating the simulations in Figs. (b) and (c). (b) Impact of under-sampling on the simulation. (c) is similar to (b), but now the under-sampling correction has been applied. In Fig 2(c), the under-sampling errors are divided by 10 at SIs ≥ 0.4 nm. Note that individual radiances simulated at different SIs are interpolated to 0.01 nm and then convolved with the Gaussian function (FWHM: 0.4 nm) which represents the OMI instrument spectral response function.








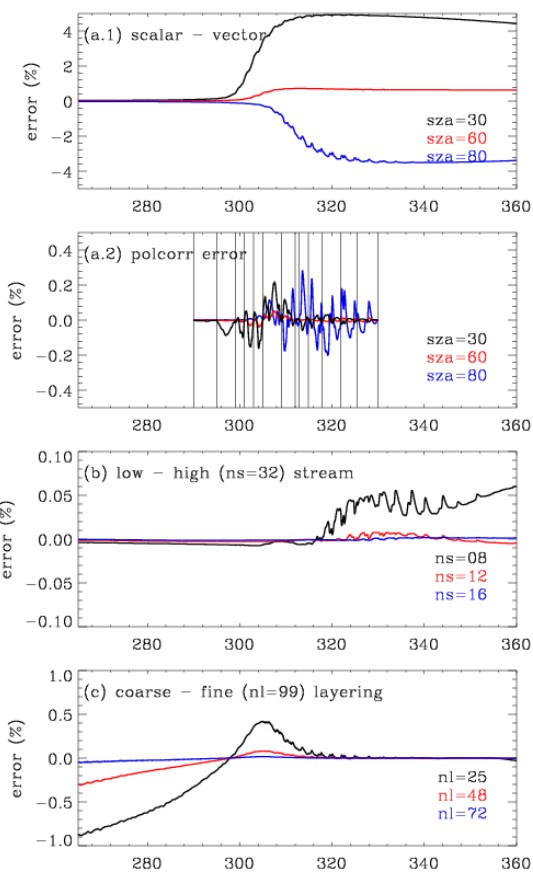


**Fig. 3.** Errors of the radiance simulation due to the RT approximation used in v1, arising from (a.1) neglecting the
polarization effect for different solar zenith angles (sza), (a.2) polarization correction errors, (b) using a low
number of streams (ns), and (c) using a coarse number of vertical grids (nl). Note that vertical lines in Fig. 3.a.2
indicate wavelengths used in deriving the on-line polarization correction spectrum.




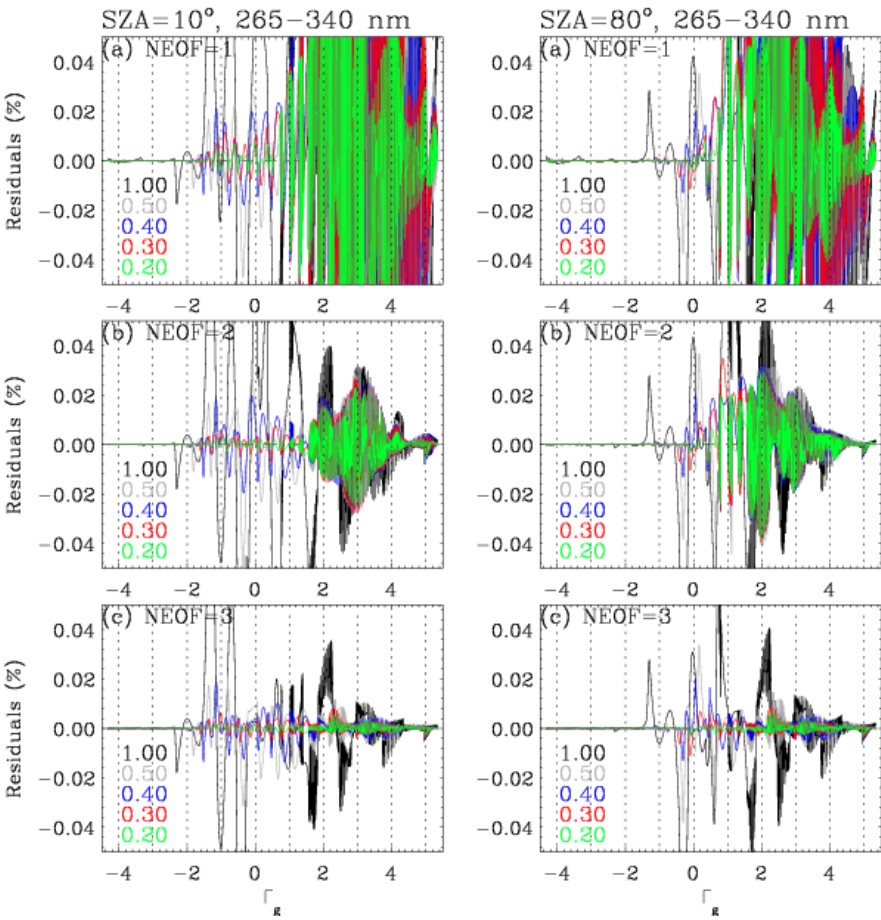


**Fig. 4**. Residuals (%) of the PCA-RT radiance in the wavelength range 265-340 nm compared to the exact-RT calculations, for different binning steps (different colors) and number of EOFs (a, b, c). Results are plotted as a function of $\Gamma_g$ (logarithm of the total gas optical depth), for solar zenith angles (SZAs) of (left)10° and (right) 80°.



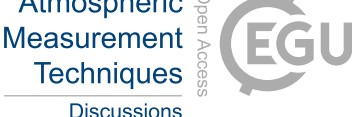

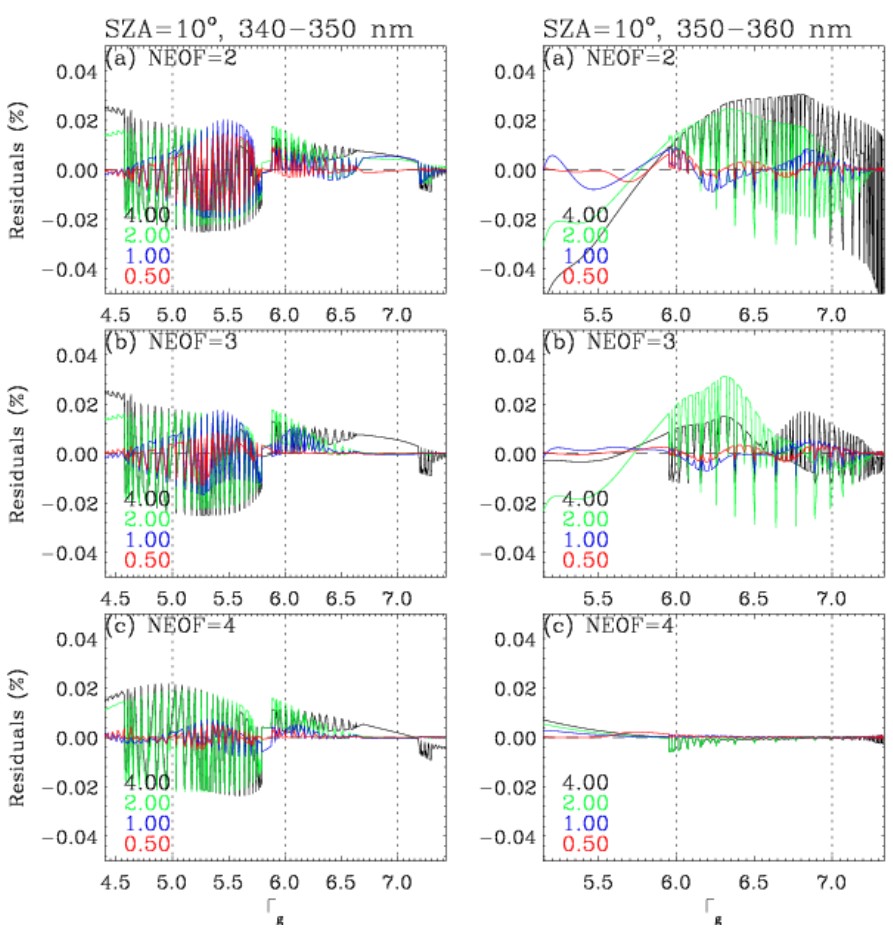


**Fig. 5.** Same as Fig. 4, but for different windows, (left) 340-350 nm and (right) 350-360 nm, respectively.








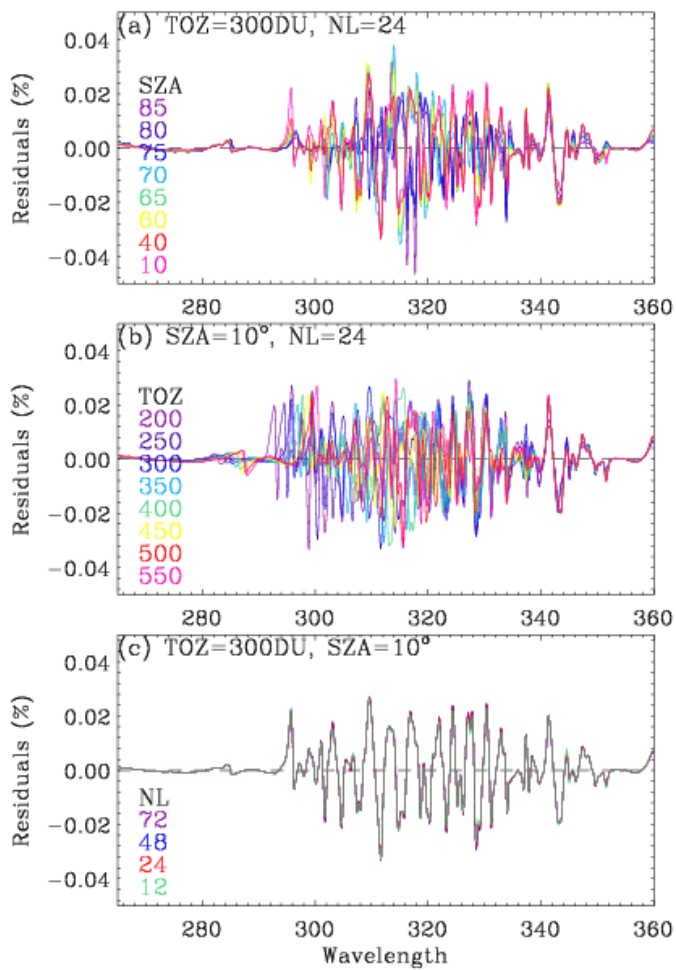


**Fig. 6**. Residuals (%) of the PCA-RT radiances with the binning scheme given in Table 1, for
various sets of (a) solar zenith angles, (b) ozone profiles with different total ozone columns
(TOZs), and (c) number of atmospheric layers.






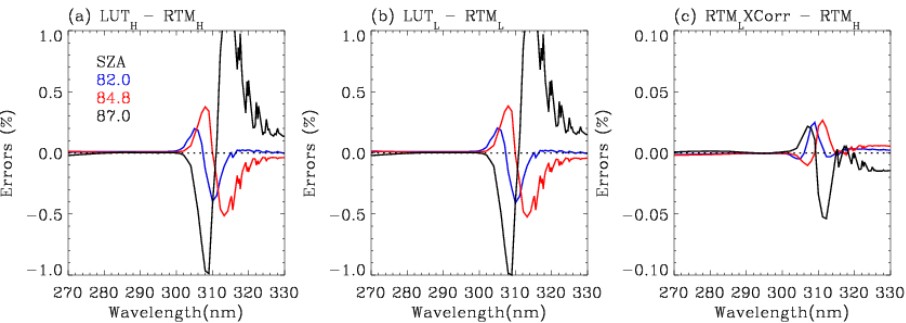


**Fig. 7.** Evaluation of simulations with respect to extreme SZAs at VZA = 61°, AZA = 0 °,
ALB=0 %, and surface pressure = 1013.25 hPa. LUT and RTM represent LUT and on-line
radiative transfer model (RTM) based calculations, respectively, with the subscripts H and L
indicating high and low accuracy configurations. $RTM_L$ X Corr radiances are simulated using
on-line RTM with low accuracy configuration, but corrected using $LUT_H/LUT_L$.

669

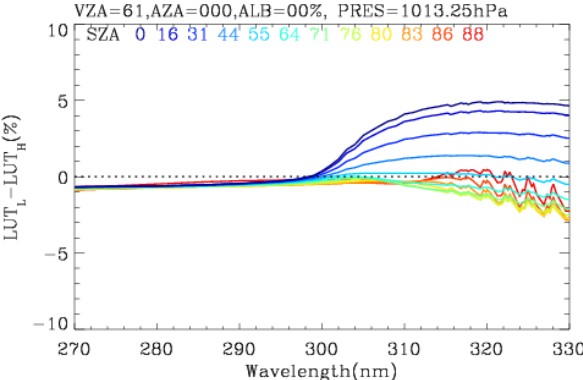

670

**Fig. 8**. Example of LUT-based correction spectrum.

672



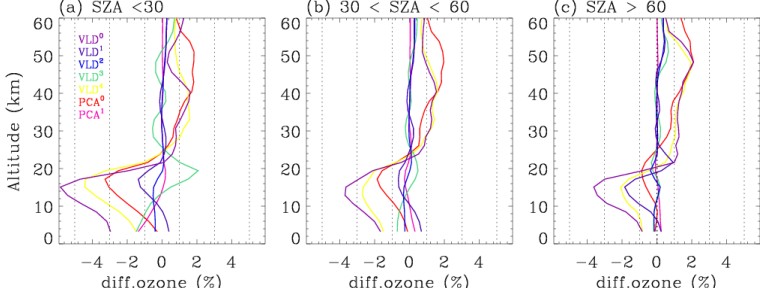

**Fig. 9**. Mean biases of ozone profile retrievals with different configurations compared to those with the reference configuration. Each configuration is given in Table 4.

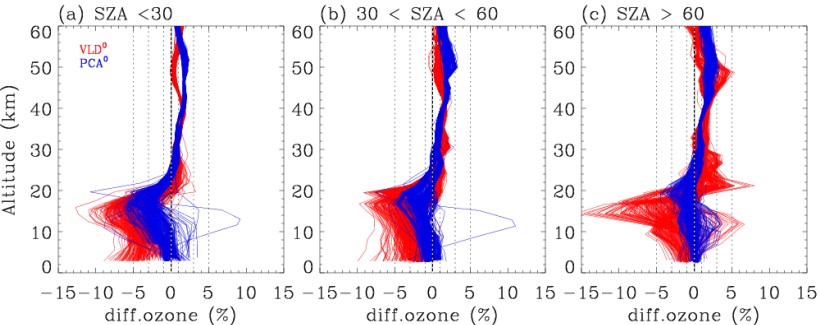

**Fig. 10.** Same as Fig.9, but for individual differences. $VLD^0$ and $PCA^0$ represent v1 and v2 forward model configurations, respectively.


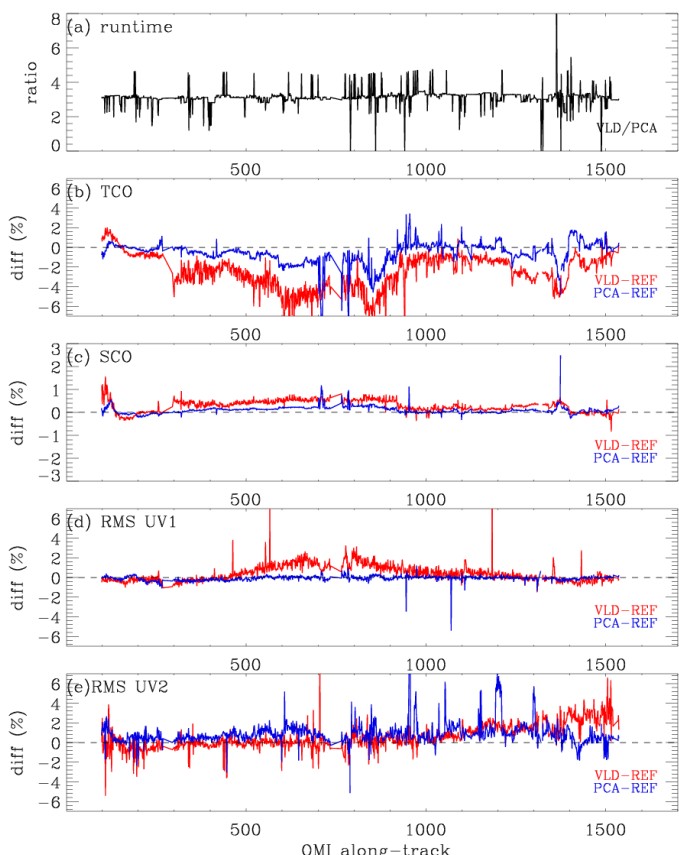

**Fig. 11.** Same as Fig. 10, but for (a) runtime, (b) tropospheric column ozone (TCO), (c)
stratospheric column ozone (SCO), and (d) UV1 (270-310 nm)/(e) UV2 (310-330 nm) fitting
residuals, along with the OMI along-track position (1-1644) at nadir cross-track. Note that the
fitting residuals are estimated as root mean square (RMS) errors for differences between
measured and simulated spectra relative to the measurement error. VLD and PCA represent
v1 and v2 forward model configurations, respectively.




