# Peer review of "Radiative transfer acceleration based on the Principal Component Analysis and Look-Up Table of corrections: Optimization and application to UV ozone profile retrievals"

_Atmospheric Measurement Techniques, 2020_

## Referee Comment (RC1) · Anonymous Referee #1 · 12 Oct 2020

This manuscript presents principal component analysis (PCA)-Look Up Table (LUT) framework to speed up the radiative transfer calculation in the remote sensing of gases using hyperspectral spectrometers. The method would benefit retrieval algorithms that need full radiance to be calculated iteratively and current space-based instrument with large data volume to be processed (like OMI and TROPOMI), especially the upcoming geostationary satellite instruments (TEMPO and GEMS). The method described in this paper is clear and the paper is generally well written.

General Comments:

[Figure]

1. The improved fast RT calculation does improve the forward model accuracy, as a consequence, the ozone profile retrieval is improved, but it is unclear about the effect of errors in the previous radiative transfer calculation (as in Fig. 3) on the released ozone profile product, https://avdc.gsfc.nasa.gov/index.php?site=1620829979&id=74. There might be systematic error that varies with ozone profile and geometries (solar zenith, viewing zenith and relative azimuth angel), and thus depends on season, latitude and cross-track position. It is helpful to give a general assessment of these errors to data users.

2. The PCA-RT and LUTs considered the Rayleigh scattering atmosphere, however, for ozone profile retrieval, in the stratosphere the effect of scattering of aerosol is small or can be account for by fitting additional 1st or 2nd-order term of albedo or cloud fraction, but for tropospheric ozone, the effect of aerosol would be significant. What is the consideration about the scattering or absorption by particles (aerosol and optical thin cloud) in the model?

Specific Comments:

1. The improved PCA-RT aims to simulate radiance to an accuracy better than 0.05%, to what extent, the ozone retrieval accuracy would be achieved?

2. In section 3.1.2, each term of EOFs would relates to the specific optical properties of scattering and absorption in the atmosphere, please explain more about: what are the 1-3 EOFs relates to?

3. One more concern is: it seems that only absorption of ozone is considered in the RT calculations, how about the effect of other trace gases like SO2, HCHO, and NO2? Do we need to apply more EOF if other gases are included, especially when there is large SO2 amount?

4. Which model is used to generate LUTH and LUTL ? please make it clear in section 3.2 .

[Figure]

Technical Corrections:

Line 124: better to use "converged" instead of "optimized"

In Fig.2b and 2c: legend (may be 0.05) in yellow are hard to see. Please change to other distinct color.

Line 147: The sentence "around 310 nm if there is no error after undersampling correction to 0.05 nm." Is hard to understand, does it mean "around 310 nm and there is no error after undersampling correction is set to 0.05 nm."?

Line 180: "simulation" should be simulate

Line 370: "Fig. c" should be Fig. 7c

Line 236ïijŽTo be more clear, "the VLIDORT and FO Q/U values" should be "Q/U values calculated by VLIDORT and FO" .

In eq. 7 and 8, what does the $\xi$ denote?
* * *

---

## Referee Comment (RC2) · Anonymous Referee #2 · 3 Nov 2020

General comments:

Accurate calculation top-of-atmosphere radiances remain essential to ozone profile retrieval techniques, but the expense of doing on-line RTM calculations at sufficient accuracy for new sensors that collect very large amounts of hyperspectral data is becoming prohibitive. Mathematical and empirical methods which improve the computational efficiency while maintaining the demanding level of accuracy required for these calculations are necessary to take advantage of new data.

[Figure]

The focus here is on particular application of such methods to the estimation of UV radiances for ozone profile retrievals. Most important details about the radiative transfer model, assumed optical properties, and tuning parameters for the EOF binning characterization are well documented. The authors present their methods in sufficient detail to to reproduce and apply them independently. The techniques described in the manuscript are thus a useful contribution to the literature. The overall quality of the manuscript is excellent so I feel the paper can be published after making some minor revisions and addressing the comments below.

1. The paper concerns improving UV ozone profile retrievals which typically require some form of soft-calibration to account for errors in either the forward model, the measurements, or both. Were all soft-calibration corrections removed from the different retrievals before producing the results presented in figures 9 and 10? If not, how can the authors compare ozone profile retrievals from a mixture of RT configurations and soft-calibrations and attribute all differences to reductions in model approximation error? Presumably one expects the soft-calibration corrections for v2 should be smaller than v1. Is this the case?

2. I suspect there is no mention of Ring effect in the paper because that is dealt with by the authors' retrieval. I think this should be mentioned at some point in the manuscript.

3. At line 272, I suspect authors don't truly mean their multiple-scattering radiance spectra are "fully accurate," but rather these spectra represent the optimally configured model setup. If this is the case please correct the wording.

4. At line 418: will reducing errors in forward model approximations actually reduce random noise error? If that is the claim, what type of random noise errors are being referred to?

5. Is there an explanation for why the UV2 fitting residuals at low latitudes in figure 11 are somewhat larger for "PCA" than "VLD," but the reverse is true for UV1?

6. The Evaluation and Summary and Conclusions sections need further editing. I suggest breaking the long spans of text into shorter, more focused paragraphs, and reconsidering the amount of technical detail. For example, is it necessary to repeat the number of wavelengths on the OMI grid or the number of times the model is executed?

Technical and grammatical comments:

Line 49 – define LUT before using abbrev.

62 – suggest using "and" or "with" rather than "+"

101 – change "work" to "perform."

107 – reword "less spectral sampling to "fewer spectral samples"

188 – remove "correspondingly"

126 – "atm" should not be superscript

126 – "for layer 0"

132 – what is an "effective wavelength"?

138 – note on VLIDORT 2.8 seems somewhat out of place. Can this be said earlier?

145 – please indicate what ozone cross-sections were used.

192 – is there a typo on the n+NN subscript on G? Also commas are missing between n and i subscript for delta and omega.

308 – the meaning of the subscript "on" is not defined in eqn. (11a).

313 – the naught subscript (0) is generally used for the solar angle, not the view angle. Is it necessary to reverse this here?

321 – does "based on" mean "at"?

334 – Use of aq1 suggests a product of two quantities, a and q1. It would be easier to

understand a single character representing this quantity. Same with aq2 here and the qr term on line 334.

337 – I believe a minus sign is missing in front of eqn. (14c).

359 – remove periods near the end of eqn. (17).

396 – change "significantly eliminated" to "significantly reduced" or just "eliminated."

411— change "in the highest" to "with the highest"

421— what is being compared to the "runtime" configuration here?

431 – despite the very respectable in performance, the word "overcome" is a bit too strong. I feel a 3x improvement speaks for itself.

441 – change "correlated" to "in"

442 – replace "bins related" with "bins within"

444 – "...PCA approximation errors for our technique"

455 – what "OMI spectral fit" does is referred to here?

588 – what does the "List" column in Table 2 represent?

604 – in table 4, please abbreviate polarization correction as "pol. corr." rather than "polcorr".

625 – parentheses missing in the legend of figure 2.

640 – please indicate what VZA and RAA are these calculations were made for?

684 – typo: change "/" to "," just before (e).

685 – this figure would be more informative if the x-axis were latitude instead of along-track number.

---

## Author Comment (AC1) · 23 Nov 2020

We would like to thank this reviewer for the constructive comments. We have tried our best to address the 2 general comments, 4 specific comments, and 7 technical comments.
* * *
**General Comments and responses**

**C 1**. The improved fast RT calculation does improve the forward model accuracy, as a consequence, the ozone profile retrieval is improved, but it is unclear about the effect of errors in the previous radiative transfer calculation (as in Fig. 3) on the released ozone profile product, https://avdc.gsfc.nasa.gov/index.php?site=1620829979&id=74. There might be systematic error that varies with ozone profile and geometries (solar zenith, viewing zenith and relative azimuth angel), and thus depends on season, latitude and cross-track position. It is helpful to give a general assessment of these errors to data users.

**R1**. As commented, OMI ozone profile retrievals are significantly biased with respect to CCD dimensions (cross-track position, and wavelengths) as well as solar zenith angles. However, in this paper, we would like to confine the scope of this subject to retrieval errors caused by forward model simulation errors by comparing retrievals using the Reference configuration. But, we are in the preparation of another companion paper to evaluate the improved ozone retrievals with respect to forward model simulation as well as other updates, through comparison with global, long-term ozonesonde dataset. In this paper, we provided the evaluation results for three solar zenith angle regimes in Figure 9, showing that the large systematic errors of~ 5- 15 % due to v1 forward model errors are greatly eliminated below 30 km.

**C2.** The PCA-RT and LUTs considered the Rayleigh scattering atmosphere, however, for ozone profile retrieval, in the stratosphere the effect of scattering of aerosol is small or can be account for by fitting additional $1^{st}$ or $2^{nd}$-order term of albedo or cloud fraction, but for tropospheric ozone, the effect of aerosol would be significant. What is the consideration about the scattering or absorption by particles (aerosol and optical thin cloud) in the model?

**R2**. The surface albedo is fitted as a first-order polynomial in UV2 (affecting tropospheric ozone retrieval) to partly account for aerosol effect and compensate for other scattering effects by clouds and surface. This seems to work reasonably well as we have not seen obvious retrieval artifacts in the presence of absorbing aerosols (e.g., Sahara dust). Our algorithm has option to include aerosols in the forward model simulation, using a mixture of six type of aerosols using monthly mean aerosol fields from model simulations. Previous tests have shown that whether to include aerosols does not significantly affect our retrievals. In addition, there are large uncertainties in aerosol optical depth inputs and it slows down the RTM (e.g., high number of streams). Therefore, we did not use this option in our retrievals.

**Responses to Specific Comments**

**C1**. The improved PCA-RT aims to simulate radiance to an accuracy better than 0.05%, to what extent, the ozone retrieval accuracy would be achieved?

**R1.** This criterion is determined to be better than measurement errors which are typically assumed

as the level of 0.1 % in the Huggins band for BUV measurements. In Figure 9, we can assess the effect of PCA-RT approximation errors (~ 0.05 % or less) on ozone retrieval errors (pink color: PCA[1]), indicating the negligible effect above 10 km and the increasing errors up to ~ 1.5 % at the bottom layer.

**C2**. In section 3.1.2, each term of EOFs would relate to the specific optical properties of scattering and absorption in the atmosphere, please explain more about: what are the 1-3 EOFs relates to?

**R2**. Starting with a strongly-correlated set of optical property profiles, and working with logarithmic quantities, the PCA process first takes the spectral mean F_0 of the data, and then performs a PCA on the mean-removed set of (logarithmic) profiles. This yields a series of EOFs ranked in order of their contributions towards capturing the variance in the original data. The PCA reshuffles the optical data, such that the mean and the 1-3 most significant EOFs will provide a much smaller set of "super-profiles" that are used as inputs to the full multiple-scatter RT calculations. The distribution of scattering and absorption layer optical thickness values in the original profiles is replaced by a different set of distributions in the smaller set of "super profiles" which are constructed from the mean and the 1-3 EOFs. It is not really possible to put a physical interpretation on the EOFs (i.e. first mode is absorption, second scattering), but one could say that the first and most important EOF contains more information about the absorption profile, since it is the trace gas absorption that provides the bulk of the variability in the original data.

**R3**. One more concern is: it seems that only absorption of ozone is considered in the RT calculations, how about the effect of other trace gases like SO2, HCHO, and NO2? Do we need to apply more EOF if other gases are included, especially when there is large SO2 amount?

**C3**. The number of EOFs required in PCA simulation depends on the spectral variation of main absorber. In our application to ozone profile retrievals with 270-330 nm, the effect of other trace gases are really week compared to ozone absorption, therefore we don't need increasing the number of EOFs.

**R4**. Which model is used to generate $LUT_H$ and $LUT_L$ ? please make it clear in section 3.2.

**C4**. In section 3.2, it is addressed like "To construct LUTs, RT calculations are performed using the VLIDORT version 2.8 model"

**Responses to Technical Corrections**

**C1**. Line 124: better to use "converged" instead of "optimized"

**R1**. It has been corrected as "converged".

**C2**. In Fig.2b and 2c: legend (may be 0.05) in yellow are hard to see. Please change to other distinct color.

**R2**. This figure has been revised to make clear.

**C3**. Line 147: The sentence "around 310 nm if there is no error after undersampling correction to 0.05 nm." Is hard to understand, does it mean "around 310 nm and there is no error after undersampling correction is set to 0.05 nm."?

**R3**. For clarification, this sentence has been revised to "Fig. 2.b illustrates that LBL calculations are required to be performed at intervals of 0.03 nm or better.

**C4**. Line 180: "simulation" should be simulate

**R4**. The associated sentence is "the LUT-based correction is applied to simulation errors". For clarification, we have corrected to "applied to approximation errors"

**C5**. Line 370: "Fig. c" should be Fig. 7c

**R5**. It has been corrected to Fig. 7c.

**C6**. Line 236:To be more clear, "the VLIDORT and FO Q/U values" should be "Q/U values calculated by VLIDORT and FO" .

**R6**. According to this comment, the relevant sentence has been revised as the differences of Q/U values calculated by VLIDORT and FO.

**C7**. In eq. 7 and 8, what does the $\xi$ denote?

**R7**. $\xi$ indicates the profile typed optical input, which has been added in the revised manuscript.

---

## Author Comment (AC2) · 23 Nov 2020

**Reponses to the second reviewer′s comments**

We would like to thank this reviewer for the constructive comments. We have addressed the 6 specific comments and 30 technical comments as shown below.
* * *
**Specific Comments and responses**

**C1**. The paper concerns improving UV ozone profile retrievals which typically require some form of soft-calibration to account for errors in either the forward model, the measurements, or both. Were all soft-calibration corrections removed from the different retrievals before producing the results presented in figures 9 and 10? If not, how can the authors compare ozone profile retrievals from a mixture of RT configurations and soft-calibrations and attribute all differences to reductions in model approximation error? Presumably one expects the soft-calibration corrections for v2 should be smaller than v1. Is this the case?

**R1**. We have not applied the soft-calibration in this paper. In this paper, v1 and v2 stand for the ozone profile retrievals where all the implementations are same, but for the forward model simulations. Actually most implementations (ozone a priori, solar reference, slit function, meteorological input, and others) are updated in both v1 and v2 here compared to those applied for generating the PROFOZ product available via AVDC. Therefore, we should re-generate the soft spectra if we want to evaluate the impact of applying different forward model simulations on ozone profile retrievals with soft calibration. We are working on updating the soft spectra, but have not yet finalized. Therefore, soft-calibration was turned off in this paper, but those will be addressed in a companion paper.

**C2**. I suspect there is no mention of Ring effect in the paper because that is dealt with by the authors' retrieval. I think this should be mentioned at some point in the manuscript.

**R2**. Yes, the ring effect is modeled using a single scattering model (Sioris and Evans, 2000) and then is fitted as a pseudo absorber. Therefore, the ring simulation is independent of the PCA-based RT simulation. and is the same between v1 and v2. This paper is intended for discussing improvements (speed and accuracy) of ozone profile retrievals with newly implemented PCA-based RT simulation and LUT-based correction. The other implementation details were referred to Liu et al. (2010) where the SAO OMI algorithm was first presented, and the updates of those will be presented in companion paper.

**C3**. At line 272, I suspect authors don't truly mean their multiple-scattering radiance spectra are "To evaluate the performance of the PCA approximation, the "exact-RT" model is executed fully accurate," but rather these spectra represent the optimally configured model setup. If this is the case, please correct the wording.

**R3**. Yes. According to Fig. 3b, 12 full stream is used to perform accurate MS calculation as the remaining errors are within 0.01%. We have revised the indicated sentence to "To evaluate the PCA approximation, the "exact-RT" model is performed where accurate full-MS VLIDORT calculations are expensively performed at every wavelength in addition to accurate single scattering calculations.

**C4**. At line 418: will reducing errors in forward model approximations actually reduce random noise error? If that is the claim, what type of random noise errors are being referred to?

**R4**. "Random-noise errors" is misprint. We intended to say that the variabilities of individual differences are reduced. For clarification, the word, ″random noise errors″ has been changed to ″the variabilities of individual differences″

**C5**. Is there an explanation for why the UV2 fitting residuals at low latitudes in figure 11 are somewhat larger for "PCA" than "VLD," but the reverse is true for UV1?

**R5**. We should be careful for evaluating the fitting residuals because the smaller fitting residuals could not directly lead to better ozone retrievals likely due to the presence of systematic measurement errors. As you indicated, using ″PCA″ results in slightly larger fitting residuals at low latitudes (Fig 11.e), but significantly improves the tropospheric column ozone retrievals at the entire latitude (Fig.11.b).

**C6**. The Evaluation and Summary and Conclusions sections need further editing. I suggest breaking the long

spans of text into shorter, more focused paragraphs, and reconsidering the amount of technical detail. For example, is it necessary to repeat the number of wavelengths on the OMI grid or the number of times the model is executed?

**R6**. Thanks for your constructive comment. We have carefully revised section 5.

**Technical Comments and responses**

**C1**. Line 49 – define LUT before using abbrev.
**R1**. Its abbreviation (Look Up Table) has been inserted.

**C2**. Line 62 – suggest using "and" or "with" rather than "+"
**R2**. Changed to "OMI and TES (Fu et al., 2013), OMI and AIRS (Fu et al., 2018), or GOME-2 and IASI (Cuesta et al., 2013)"

**C3**. Line 101 – change "work" to "perform."
**R3**. The related sentence is "the PCA model is employed in this work ". We think that "work" should be kept as it is used as a noun.

**C4**. Line 107 – reword "less spectral sampling to "fewer spectral samples"
**R4**. This suggestion is accepted in the revised manuscript.

**C5**. Line 188 – remove "correspondingly"
**R5**. This suggestion is accepted in the revised manuscript.

**C6**. Line 126 – "atm" should not be superscript 126 – "for layer 0"
**R6**. $P_i = 2^{-\frac{i}{2}}\,atm$ has been revised to $P_i = 2^{-\frac{i}{2}}$ (in atm, 1 atm = 1013.25 hPa)

**C7**. Line 132 – what is an "effective wavelength"?
**R7**. For clarification, "effective" is deleted.

**C8**. Line 138 – note on VLIDORT 2.8 seems somewhat out of place. Can this be said earlier?
**R8**. To address this comment, we have put "The Vector Linearized Discrete Ordinate Radiative Transfer (VLIDORT) model v2.4 (Spurr et al., 2006) was employed as a forward model in the v1 OMI ozone profile algorithm (Liu et al., 2010) implemented at SIPS. We have updated VLIDORT to the latest version v2.8 for this study as well as in the PCA-VLIDORT described in Sect. 3. Note that there is little difference between v2.4 and v2.8 in term of simulation accuracy." before " The simulation is iteratively…" in the first paragraph of Section 2.

**C9**. Line 145 – please indicate what ozone cross-sections were used.
**R9**. The related sentence has been revised as "at the sampling rate (0.01 nm) of the ozone cross sections (Brion et al., 1993)"

**C10**. Line 192 – is there a typo on the n+NN subscript on G? Also commas are missing between n and i subscript for delta and omega.
**R10**. $G_{n+N_N,i}$ has been corrected to $G_{n+N_L,i}$. The comma has been inserted between subscripts.

**C11**. Line 308 – the meaning of the subscript "on" is not defined in eqn. (11a).
**R11**. The subscript "on" indicates the on-line calculation, whereas the subscript "LUT" indicates the LUT-based calculation. For clarification, we have inserted "where the subscripts "on" and "LUT" stand for on-line and LUT-based calculations, respectively" before the equation 11.

**C12**. Line 313 – the naught subscript (0) is generally used for the solar angle, not the view angle. Is it

necessary to reverse this here?
**R12**. The manuscript has been revised to use the subscript 0 for solar zenith angle.

**C13**. Line 321 – does "based on" mean "at"?
**R13**. This suggestion has been accepted.

**C14**. Line 334 – Use of aq1 suggests a product of two quantities, a and q1. It would be easier to C3 AMTD Interactive comment Printer-friendly version Discussion paper understand a single character representing this quantity. Same with aq2 here and the qr term on line 334.
**R14**. According to this suggestion, we have changed the symbols aq1 and aq2 to k1 and k2.

**C15**. Line 337 – I believe a minus sign is missing in front of eqn. (14c).
**R15**. the minus sign has been inserted.

**C16**. Line 359 – remove periods near the end of eqn. (17).
**C16**. Yes, this equation has been revised.

**C17**. Line 396 – change "significantly eliminated" to "significantly reduced" or just "eliminated."
**R17**. It has been changed to ″significantly reduced″.

**C18.** Line 411 change "in the highest" to "with the highest"
**R18**. It has been changed to ″"with the highest"

**C19**. Line 421 what is being compared to the "runtime" configuration here? ˇ
**R19**. This runtime represents the algorithm running time taken to retrieve individual pixels, which are evaluated with different forward models in Fig. 11.a. So the runtime of v2 is compared to the runtime of v1 here. For clarification, we changed the sentence to "the comparison of the runtime (Fig 11.a) demonstrates that v2 is faster by a factor of 3.3 on average"

**C20**. Line 431 despite the very respectable in performance, the word "overcome" is a bit too strong. I feel an improvement speaks for itself.
**R20**. We agreed with that. "Overcome" has been changed to "improve"

**C21**. Line 441 change "correlated" to "in"
**R21**. This suggestion has been accepted.

**C22.** Line 442 replace "bins related" with "bins within"
**R22**.This suggestion has been accepted.

**C23** Line 444 ". . .PCA approximation errors for our technique"
**R23**. We have edited them to "PCA approximation errors for our application"

**C24** Line 455 what "OMI spectral fit" does is referred to here?
**R24**.This one represents the v1 retrieval of OMI ozone profile, the associated sentence has been deleted during the revision.

**C25** Line 588 what does the "List" column in Table 2 represent?
**R25**. It represents the order of bins. For clarification, it has been deleted.

**C26**. Line 604 in table 4, please abbreviate polarization correction as "pol. corr." rather than "polcorr".
**R26**. This suggestion has been accepted.

**C27.** Line 625 parentheses missing in the legend of figure 2.
**R27**. Thanks for your careful review, this figure has been revised.

**C28**. Line 640 please indicate what VZA and RAA are these calculations were made for?
**R28**. We have revised all captions to indicate viewing geometries performed for each testing.

**C29**. Line 684 typo: change "/" to "," just before (e).
**R29**. "/" has been changed to "and".

**C30** Line 685 this figure would be more informative if the x-axis were latitude instead of along track number.
**R30**. This figure has been re-plotted in the axis based on the latitude.